# Impact of Toceranib Phosphate and Carprofen on Survival and Quality of Life in Dogs with Inflammatory Mammary Carcinomas

**DOI:** 10.3390/vetsci11090430

**Published:** 2024-09-13

**Authors:** Miguel Garcia-de la Virgen, Isabel Del Portillo Miguel, Elisa Maiques, Ignacio Pérez Roger, Enric Poch, Juan Borrego

**Affiliations:** 1Oncology Service, Hospital Aúna Especialidades Veterinarias, IVC Evidensia, 46980 Paterna, Valencia, Spain; 2The Hospital for Small Animals, The Royal (Dick) School of Veterinary Studies, The University of Edinburgh, Edinburgh EH8 9YL, UK; idelpor@ed.ac.uk; 3Department of Biomedical Sciences, School of Health Sciences, Universidad Cardenal Herrera-CEU, CEU Universities, 46115 Alfara del Patriarca, Valencia, Spain; emaiques@uchceu.es (E.M.); iperez@uchceu.es (I.P.R.); epoch@uchceu.es (E.P.)

**Keywords:** toceranib, canine cancer, quality of life, COX-2 inhibitors, inflammatory mammary cancer, chemotherapy

## Abstract

**Simple Summary:**

Canine inflammatory mammary carcinoma is an aggressive and painful type of mammary gland cancer in dogs, often leading to poor quality of life and short survival times. This study investigated the effectiveness of a treatment combining two drugs, toceranib and carprofen, as a first-line therapy for 15 dogs diagnosed with this condition. The aim was to evaluate both the cancer’s response to the treatment and any improvements in the dogs’ quality of life. The results showed that while no measurable responses were observed, 60% of the dogs experienced stabilization of the disease. Importantly, there were significant improvements in the dogs’ overall well-being, including reduced pain and better mobility, appetite, and happiness. The median progression-free survival and the overall survival time were of 76 and 90 days, respectively, which is a notable outcome for such a severe condition. These findings suggest that this drug combination can provide meaningful relief and extend life for dogs suffering from inflammatory mammary carcinoma, offering hope and better management options for pet owners and veterinarians dealing with this challenging disease.

**Abstract:**

Canine inflammatory mammary carcinoma (IMC) is an aggressive and rare type of mammary gland cancer in dogs where vascular endothelial growth factor and cyclooxigenase-2 overexpression usually occur, which contribute to its invasive and angiogenic nature. This study aimed to evaluate the efficacy and safety of a combined treatment regimen of toceranib phosphate and carprofen in dogs with measurable IMC. Fifteen female dogs with histopathologically confirmed IMC were included, undergoing a regimen of toceranib (2.4–2.75 mg/kg PO, three times weekly) and carprofen (4.4 mg/kg/24 h PO). Initial evaluations included physical exams, tumor measurements, complete blood count, biochemistry, urinalysis, three view thoracic radiographs, and abdominal ultrasound. Follow-up assessments of physical condition and quality of life (QOL) were conducted bi-weekly, with tumor response evaluations monthly, using RECIST v1.0 criteria. While no complete or partial responses were observed, 60% of the dogs maintained stable disease, with a median progression-free survival of 76 days and an overall survival of 90 days. Notably, 60% of the dogs showed clinical benefit through improved QOL and disease stabilization. The treatment was well-tolerated, with only grade I/II toxicities reported. Despite limited biological activity against the cancer, this protocol may enhance QOL in dogs with IMC, offering a valuable palliative option.

## 1. Introduction

Canine inflammatory mammary carcinoma (IMC) represents an infrequent type of mammary tumors, accounting for 7.6% of all mammary tumors diagnosed in female dogs [1]. This condition has also been described in isolated cases in male dogs and cats [2,3]. Clinically canine IMC is characterized by the presence of edema, swelling, warmth, and pain affecting the entire mammary chain. Most cases rapidly develop distant metastasis or systemic clinical signs [4,5]. Canine IMC exhibits an aggressive biological behavior, resulting in a poor prognosis, with patients achieving median survival times around 1–2 months [4,6].

In women, inflammatory breast cancer (IBC) faces a poor prognosis similar to that of dogs; however, long-term survivors have been reported following multimodal treatments involving surgery, radiotherapy and chemotherapy [7,8]. In dogs, surgery is typically not indicated due to extensive skin involvement, the presence of local coagulopathies, and metastasis at diagnosis. Consequently, surgical candidates are selected on a case-by-case basis [4]. Furthermore, traditional chemotherapy using doxorubicin has not demonstrated any benefit in local control or metastasis prevention [9,10]. Recent assessments of palliative radiation therapy combined with toceranib, thalidomide, and piroxicam have shown prolonged survival time and time to progression in these patients, though further studies are required to confirm the benefits of this approach [11].

It is well established that several types of cancer overexpress cyclooxygenase-2 (COX-2) and cyclooxygenase-1 (COX-1), which has driven the use of non-steroidal anti-inflammatory drugs (NSAIDs) in oncology [12]. Specifically, carprofen has demonstrated cytotoxic activity against osteosarcoma cells in vitro [13], although no in vivo studies have been published. Carprofen, a COX-2 selective NSAID, plays a crucial role in inflammation and provides adequate analgesia, which can benefit patients with IMC. Potential side effects of NSAID use include gastrointestinal tract ulceration, renal damage, hepatopathy, and inhibition of platelet function [14].

Toceranib is a multitarget tyrosine kinase inhibitor (TKI), similar to sunitinib, with activity against proto-oncogene receptor tyrosine kinase KIT, vascular endothelial growth factor receptor-2 (VEGFR2), and platelet-derived growth factor receptors (PDGFR α and β), as well as the capacity to decrease regulatory T cells [15]. This drug was initially licensed for recurrent mast cell tumors [16]; however, several different solid tumors have also been shown to overexpress various tyrosine kinase proteins. Therefore, it is not surprising that toceranib has been investigated in other types of cancer, such as neuroendocrine tumors and various solid carcinomas including mammary, among others [17]. Given the observed expression of VEGF or aberrant VEGF signaling in canine IMC [18,19,20], the use of toceranib in this disease seems reasonable. Another recent study has looked at the combination of toceranib phosphate and cyclophosphamide together with NSAIDs in dogs with IMC showing good tolerance and promising results [21].

In veterinary medicine, the visual analog scale (VAS) and quality of life (QOL) questionnaires are pivotal instruments for evaluating the well-being of canine cancer patients, particularly in palliative care settings. These tools offer subjective yet quantifiable assessments of a pet’s comfort, pain levels, and overall QOL from both pet owners’ and veterinarians’ perspectives. The VAS provides a straightforward method for evaluating pain and discomfort on a scale, while QOL questionnaires encompass broader aspects of a pet’s daily life, including physical functioning, emotional well-being, and social interactions. This comprehensive assessment is crucial in specific cancer types like canine IMC that usually receive palliative care treatment, where the primary objective is to alleviate suffering and maintain the highest possible QOL for the animal [22,23].

The aim of this study was to assess the antitumoral efficacy and toxicity profile of the combination of carprofen and toceranib, as well as the impact on the QOL of canine IMC cases after receiving toceranib phosphate and carprofen.

## 2. Materials and Methods

### 2.1. Patient Selection

The study retrospectively included consecutive female dogs that met the inclusion criteria, which were presented to the Instituto Veterinario de Oncología Comparada (IVOC) between December 2012 and December 2015, diagnosed with inflammatory mammary carcinoma (IMC), and treated with toceranib and carprofen. A clinical diagnosis of IMC was made after identifying typical clinical signs (erythema, edema) alongside a histopathological diagnosis. The type of incisional biopsy was chosen based on the clinician’s preference. The two types of incisional biopsies performed were wedge biopsy, obtaining at least two tissue fragments with a scalpel, or punch biopsy, using a commercial punch of 4–6 mm in diameter, taking at least two samples. In both cases, tissue including skin was obtained, and the cutaneous defect was sutured using a 2/0 or 3/0 non-absorbable monofilament suture. All cases were sedated or anesthetized for this procedure.

All samples obtained were fixed in 10% neutral buffered formalin for 24–72 h and sent for histopathological analysis. Tissue samples were routinely processed, obtaining a paraffin block. Sections were sectioned (3–5 µm) for staining with hematoxylin and eosin. The detection of neoplastic emboli in the dermal lymphatic vessels was mandatory to confirm the diagnosis of IMC [5].

Dogs with lesions that developed within the normal mammary glands were classified as primary IMC, while dogs that developed the disease after previous surgical removal and diagnosis of non-inflammatory mammary carcinoma were classified as secondary.

### 2.2. Initial Evaluation

Initial evaluation required a physical examination (PE), including measurement of the longest diameter of the primary lesion with a caliper, an abdominal ultrasound (AUS), and three-view thoracic radiographs. Complete blood count (CBC), biochemistry (including at least alanine aminotransferase (ALT), alkaline phosphatase (ALKP), creatinine, urea, glucose, and total protein), and baseline urinalysis (which included at least pH, density, glucose, ketone bodies, nitrites, leukocytes, bilirubin, and total proteins) were mandatory. These tests were performed at least seven days prior to the start of the treatment.

In cases where the regional lymph nodes (inguinal or axillary), abdominal lymph nodes, or other organs were found to be enlarged or abnormal, a fine needle aspirate (FNA) and cytology were performed using a 23–25 G needle. Samples were spread on a slide and stained with quick panoptic stain (Hemacolor, Merck, Darmstadt, Germany). The presence of one atypical epithelial cell or atypical epithelial cell clusters with nuclear atypia within the normal lymphoid population was considered sufficient to confirm lymph node metastasis.

### 2.3. Inclusion and Exclusion Criteria

After the first consultation, dogs were eligible for the study if they met the following inclusion criteria:Presence of macroscopic lesions measurable according to response evaluation criteria in solid tumors RECIST criteria [24].Hematological and biochemical parameters less than grade 2 following VCOG-CTCAE [25].Previous use of NSAIDs and prednisolone was allowed with a washout period of 1 week before being included in the study.The use of other analgesic treatments was accepted as long as the patient was already receiving them for at least 5 days before starting the study.Dogs with regional or distant metastases were allowed to enter the study.An owner’s written consent was obtained in all cases, along with a document providing study information, QOL questionnaire document to be filled out every 2 weeks, and a document outlining the protocol to be followed.

Dogs were excluded from the analysis if they met any of the following exclusion criteria:Prior treatment with chemotherapy or tyrosine kinase inhibitors.Diagnosis of other forms of mammary gland tumors.

### 2.4. Treatment Protocol

Carprofen (Rimadyl^®^ Pfizer Animal Health Inc., Parsippany, NJ, USA) was prescribed at 4.4 mg/kg orally once daily, administered with food. The tablets could be split to achieve the nearest recommended dose.

Toceranib phosphate (Palladia^®^ Pfizer Animal Health Inc., Parsippany, NJ, USA) was administered orally three times a week (Monday, Wednesday, and Friday) at a dose ranging between 2.4–2.7 mg/kg based on preliminary data [26]. It was always administered with gloves and without splitting the tablets.

### 2.5. Data Collection

Clinical information included age, sex, breed, body weight, and neuter status (including sterilization dates and previous hormonal treatments, if any). Additionally, the location and size of the tumor, previous treatments and surgeries, type of presentation (primary or secondary IMC), histopathology results, type of biopsy, and associated complications were recorded. The response following RECIST criteria and the response of non-target lesions not included in the RECIST system (such as lymphedema, redness, firmness, and pain) were also noted. Clinical improvement was assessed based on questionnaires of QOL completed by the owner. Toxicities were evaluated according to VCOG-CTCAE criteria, and information on dose reductions, treatment delays, rescue treatments, cause of death, and necropsy findings (if performed) was included.

### 2.6. Treatment Response and Toxicity Assessment

Fourteen days after starting the treatment, a complete physical examination and a CBC were performed, alongside a QOL questionnaire filled out by the owner. One month after starting the treatment, the initial evaluation was repeated and subsequently conducted on a monthly basis. Treatment was continued as long as the tumor was responding or at least stabilized.

Response was defined according to RECIST criteria: complete remission (CR), partial remission (PR), stable disease (SD, which for this study was required to last at least one month), and progressive disease (PD). Clinical benefit (CB) was defined as the sum of all dogs achieving CR, PR, or SD for at least one month with an improvement in their QOL. This was assessed by a VAS completed by the owners, as well as a modified QOL questionnaire previously described by Lynch et al. [23] (Appendix A). The most relevant aspects were assessed, including happiness, mental state, pain, appetite, hygiene, hydration, and mobility. Side effects were graded using the VCOG-CTCAE criteria for adverse events (AEs).

### 2.7. Statistical Analysis

Statistical analysis was performed using the commercial software GraphPad Prism 7.0 (San Diego, CA, USA). The Kaplan–Meier product-limit method was used to determine median progression-free survival (PFS) and median overall survival (OS). PFS was defined as the time from inclusion to the study (starting treatment) to disease progression, while OS was defined as the time from inclusion in the study to death.

For the evaluation of the QOL questionnaires, instead of using a paired t-test, which assumes a Gaussian distribution, the Wilcoxon signed-rank test was used for the analysis. The means of the group of dogs at diagnosis were compared at days 15 and 30. Significance was set at *p* ≤ 0.05.

## 3. Results

### 3.1. Dogs and Tumor Characteristics

Eighteen dogs were initially evaluated, but only fifteen cases met the inclusion criteria. Three cases were not enrolled as they did not present measurable lesions. The main characteristics of the included cases are summarized in Table 1.

The median age was 11.6 years (range, 8–16 years) and the median weight was 21.8 kg (range, 3.3–42.7 kg) at presentation. There were six mixed-breed dogs (40%), two Yorkshire terriers (13%), and one (6.6%) each of the following breeds: French bulldog, Labrador, rottweiler, English cocker spaniel, German shepherd, and boxer. Seven female dogs were spayed (46.6%), with the median number of years since spaying being 6.5 years. Twelve dogs (80%) were diagnosed with primary IMC, while three (20%) had secondary IMC with a previous diagnosis of different non-inflammatory carcinoma. Two dogs presented with a recurrence of IMC after previous surgeries performed at different centers. None of the dogs received prior hormonal therapies.

The mean size of the longest diameter of the tumors at diagnosis was 5.79 cm. Nine dogs (60%) had lesions affecting the inguinal mammary glands, and six dogs (40%) had lesions in the axillary glands. In nine dogs (60%), the tumors affected at least two mammary glands, while in six cases (40%) only one mammary gland was affected. Erythema, warmth, and firmness were present in all cases (100%). The most common clinical signs included the presence of cutaneous nodules in 60% of cases, regional lymphadenopathy (53.4%), lymphedema of the hindlimbs (26.6%), and ulceration of the mass (13.3%) (Figure 1).

In thirteen cases, the histopathological confirmation was obtained by an incisional biopsy, while two cases were presented as a recurrence of a previous IMC with available histopathology reports. All cases showed the typical histopathological pattern of dermal involvement described in IMC (Figure 2). Samples were obtained by punch biopsy except in three cases where wedge biopsy was performed. The three dogs that underwent wedge biopsy presented complications such as wound dehiscence, and one case experienced massive bleeding requiring hospitalization and blood product administration; fortunately, the dog was discharged within 24 h. An infection developed at the biopsy site in one dog after a punch biopsy, which resolved completely with supportive care.

The median hematocrit at presentation was 41.2%, with four patients presenting mild anemia (26.6%) and eleven cases (73.3%) presenting leukocytosis. Five cases (33%) showed increases in ALKP, with four dogs having received corticosteroids. Two patients presented increases in cholesterol and one in ALT.

Lymph node metastasis was diagnosed by cytology in four cases affecting the regional lymph nodes (26.6%) and the abdominal iliac lymph nodes in four other cases (26.6%). Distant metastases were detected in six dogs (40%), with one case presenting an aggressive bone lesion in the tibia, which cytology confirmed as metastasis of carcinoma (Figure 3). No lung metastasis was detected in any case. Only one dog had a slightly prominent sternal lymph node; however, FNA cytology results revealed the presence of a reactive lymph node.

### 3.2. Treatment and Adverse Events

None of the dogs received chemotherapy treatment prior to the study. Three dogs received NSAIDs (two firocoxib and one meloxicam), and six dogs received corticosteroid therapy (one with dexamethasone, one with methylprednisolone, and four with prednisone). Five dogs were receiving tramadol, which was continued during the treatment.

The median dose of toceranib phosphate used was 2.61 mg/kg, administered orally on a Monday, Wednesday, and Friday schedule. The median dose of carprofen was 4.25 mg/kg, administered orally once per day.

AEs were mild and not dose-limiting. Six dogs missed a dose of toceranib on one occasion due to either owner oversight (two cases) or owners waiting to contact the hospital for advice after developing side effects.

Sixty percent of the dogs experienced a grade 1 or 2 AE; however, no grade 3 or 4 AEs were observed. The most frequent toxicities were gastrointestinal (anorexia, vomiting, and diarrhea), accounting for 80% of the cases. Hematological and cutaneous toxicities (hypopigmentation) were each experienced by one dog.

### 3.3. Treatment Response

No complete or partial response was recorded in any of the cases at day 30. Nine dogs (60%) maintained SD, while the remaining cases (40%) exhibited PD.

Lymphedema improved in 2 of the 4 cases (50%), disappearing in the remaining two dogs. Improvement in signs of pain, erythema, and firmness of the lesions was observed in at least one variable in 12 dogs (80%) at day 15 and in 10 cases (66%) at day 30 compared to the beginning of the study (Figure 4).

The median PFS was 76 days, and the median OS time was 90 days (Figure 5). Nine dogs (60%) experienced CB, including an improvement in their QOL.

### 3.4. Quality of Life Assessment

Changes in QOL were analyzed at the time of inclusion in the study, at day 15, and at day 30. The VAS showed a significant increase in QOL between the time of diagnosis (mean VAS 3.6) and day 15 (mean VAS 6.38) (*p* = 0.0009), as well as between the day of diagnosis and day 30 (mean VAS 6.75) (*p* = 0.001). However, there were no significant differences between day 15 and day 30 (*p* = 0.22) (Figure 6).

Regarding the questionnaire, happiness, mental state, appetite, and general mobility showed statistically significant improvements compared to the beginning of the study. Specifically, one month after treatment, clients reported that dogs were playing more (*p* = 0.049), having better days than worse (*p* = 0.0059), experiencing less pain (*p* = 0.0020), eating more of their usual food (*p* = 0.0082), moving more easily (*p* = 0.0215), and being more active than before (*p* = 0.0039) compared to the time of diagnosis. Owners also described their pets as having better general health at day 15 (*p* = 0.0001) and day 30 (*p* = 0.0005) post-diagnosis (Table 2).

### 3.5. Follow-Up and Rescue Treatments

Rescue treatments were administered only after PD. Five cases received rescue treatment with doxorubicin at 30 mg/m^2^ by slow intravenous (IV) infusion every three weeks, while in four cases, no further cytotoxic treatments were chosen. Only one patient that started doxorubicin received more than one dose. In all cases, the analgesia schedule was modified by combining tramadol, NSAIDs, and paracetamol/codeine. Three cases also received corticosteroids.

No patients were lost to follow-up. All cases died due to progressive disease and clinical signs associated with IMC, with three cases developing lung metastases. Thirteen cases were euthanized, while two died at home after owners refused euthanasia due to the worsening of their QOL. Necropsy was performed on only one dog, confirming metastases in the lymph nodes and lungs.

## 4. Discussion

Canine IMC is one of the most aggressive mammary cancers in dogs, being incurable and thus necessitating the development of anticancer strategies to improve prognosis [4,8,9]. This neoplasm, similar to women IBC, exhibits a rapid-growing, angioinvasive, and angiogenic phenotype [20]. The rationale behind using toceranib phosphate in our study was based on that phenotype and some of the therapeutic approaches for aggressive mammary cancer in human medicine focusing on inhibiting VEGF and angiogenesis, which are essential in carcinogenesis [27]. Deregulation of VEGFR-2 expression promotes vasculogenic mimicry (VM), which correlates with poor prognosis in canine mammary gland tumors (CMT). Antiangiogenic agents like sorafenib, a VEGFR-2 inhibitor, have shown efficacy in inhibiting VM in CMT, highlighting their potential in improving outcomes for these patients using that therapeutic approach [28]. This has provided an improvement in the disease-free interval but rarely an increase in the overall survival time in human patients [29].

Our study introduced a dual-drug regimen using Palladia and carprofen for treating IMC in dogs. This approach aimed not only to improve clinical outcomes but also to expand treatment options in a palliative setting, offering a viable and cost-effective alternative to protocols that include radiotherapy and certain intravenous chemotherapy treatments.

Our study included only 15 dogs and lacked a control group mainly due to the low incidence of the disease and the absence of a standard treatment for canine IMC. As a result, our outcomes could only be compared with other studies, which have inherent biases due to different schemes and treatment protocols. We had the same veterinarian documented clinical findings and treatment responses, ensuring complete records and thereby diminishing information bias common in some studies.

The median age of the patients in this study was similar to that reported in the literature, being all geriatric patients and usually more than 10 years old [9,11,21]. In addition, several studies have reported a higher incidence of canine IMC in large breeds [3,4], and this was reinforced in this study with a median weight of more than 20 kg.

One of the few prognostic factors described in canine IMC is the classification between primary or secondary, where the primary appears to be associated with a worse prognosis [30]. In this study, the higher proportion of primary canine IMC was similar to other studies [4,9], although this distribution has not been consistent in other studies [1,10,30]. In this study, primary canine IMC was not associated with worse prognosis, and survival times were identical in both primary and secondary IMC. The lack of statistical significance could be related to the small sample size of the study compared to previous studies [30].

Almost half of the patients in this study were spayed (46.6%), and none received previous hormonal therapies in contrast to previous studies [9,30]. However, most of the patients were spayed after the first heat. The location and macroscopic appearance were similar to previous studies, with the inguinal mammary glands being the most commonly affected and erythema and warmth the typical appearance, followed by lymphedema in hindlimbs, regional lymphadenopathy, and ulceration [4,30]. The majority of our patients presented with nodular lesions, which are easier to measure using RECIST criteria, differing from the study by Alonso-Miguel et al., which included more plaque-like lesions potentially being associated with a better outcome than nodular lesions [21].

Most published studies on canine IMC have included cases where RECIST criteria were inapplicable due to unmeasurable lesions. Similarly to humans, most canine IMC patients present with infiltrating tumors rather than measurable masses, making conventional measurement and follow-up challenging. In human medicine, it has been suggested that tumor size does not have the same prognostic value in IBC patients compared to those with non-IBC [31]. Consequently, response assessments were based on subjective variables, introducing several inherent biases [4,10]. To address this, our study included only measurable lesions to objectively assess the response, excluding three cases with unmeasurable lesions at the initial evaluation. Due to these limitations of the RECIST system [24] and in studies involving antiangiogenic drugs [32,33], our study included measurable tumors using these criteria combined with QOL assessments through questionnaires to evaluate the biological efficacy of the treatment. Despite not achieving responses according to RECIST criteria [23], our study observed a comparable rate of SD (60%) compared to the 43.8% reported in the only prospective article assessing toceranib, piroxicam, and thalidomide [11]. A recent study assessing a combination of toceranib, different NSAIDs, and cyclophosphamide reported a stable disease rate of 66%, although only three cases were eligible for RECIST criteria, with two maintaining stable disease [21]. Previous studies have indicated that patients with this disease typically live no more than 30–60 days post-diagnosis [1,4,9]. Although comparison with previous studies is complicated due to various limitations, including study designs and different treatments, our study’s stable disease rate of 60% using objective criteria (RECIST) is similar than in previous studies [11,21]. Overrepresentation of this type of macroscopic measurable tumors could have created a selection bias.

Canine IMC has been described to have a different pattern of metastasis compared to other mammary tumors, with the bladder and sexual organs being affected [9]. In our cases, metastasis was only reported in regional or abdominal lymph nodes as in other studies [1,4], and one case was diagnosed with a metastatic bone lesion in the diaphysis of the tibia based on FNA and cytology. Although this has not been frequently reported in dogs, it is a normal finding in humans [34].

In all cases where a wedge biopsy was performed, complications were detected, while those performed by punch biopsy had only one patient developing complications (8.3%), suggesting that the latter technique might be safer while obtaining a biopsy in canine IMC. This was also observed in the studies by Clemente et al., where no major complications were described when performing punch biopsies [9]. One dog experienced severe bleeding after a wedge biopsy, which could be related to abnormalities in coagulation as previously described with canine IMC (21%) [4,35]. In this study, a coagulation profile was not performed as in most of the previous publications [1,9,10,11,21,30], but due to the severity of this potential complication, it is recommended to assess a coagulation profile if a biopsy is required.

Leukocytosis at initial presentation was the most common finding in the bloodwork of the patients, achieving normal limits one month after starting treatment in 45% (5/11) of the cases. This can reflect the impact of the treatment in controlling the systemic inflammatory component of canine IMC.

The medical treatment of dogs with IMC is largely palliative due to its aggressiveness. Previous studies describe the use of palliative treatments, including the administration of NSAIDs, analgesics, or steroidal anti-inflammatory drugs [4,9,10]. The benefit of these treatments is the analgesic effect of the drug as well as, in the case of NSAIDs, their potential antitumor activity based on COX-2 inhibition, which is an essential component of carcinogenesis and is overexpressed in IMC [36,37]. It has been previously shown that anti-COX-2 drugs appear to increase survival in canine IMC [9,36]. In another study, all dogs included with IMC receiving piroxicam as a sole agent experienced a decrease in erythema, edema, and pain, as well as an improvement in their mobility and appetite [10]. In our study, carprofen was selected due to its selective inhibition of COX-2 and its low incidence of gastrointestinal and renal effects [38].

Traditional chemotherapy has been attempted in the management of canine IMC, showing mild improvement in the median OS of these patients compared to NSAIDs; however, the results were not statistically significant [9]. Toceranib phosphate has been previously used in two studies in combination with other metronomic drugs and radiation therapy (RT) [11,21]. Rossi et al. evaluated the concomitant administration of toceranib with thalidomide, with or without palliative RT [11]. This prospective study showed a SD rate of 43% in the group treated medically and 100% in the group treated with palliative RT (4 cases). In contrast to our study, these patients received thalidomide, which could have a synergistic effect due to its anti-angiogenic actions and immune-mediated effects. The most recent study assessing toceranib combined with cyclophosphamide showed a CB of 100%, although this study was retrospective and only three patients were eligible for RECIST criteria [21]. In our study, the sample size was higher, with 15 cases evaluated via RECIST criteria. Nine dogs (60%) met the criteria to be included in the definition of CB, which included an improvement in VAS in the QOL questionnaire and CR, PR or SD according to RECIST criteria maintained for at least one month.

The median OS in our study was 90 days, higher than some studies [4,9] but less than a study assessing the use of piroxicam that achieved a median overall survival time of 183 days [10]. The difference in our study is that we followed RECIST criteria to evaluate [23] the response providing the mean tumor measurements. Furthermore, these differences may be due not only to different treatments and designs used but also to the small sample size. Moreover, the median survival time achieved in our patients was higher than the survival achieved in the most recent prospective study, with a median of 59 days in the group treated with toceranib, thalidomide, and piroxicam. Nevertheless, our OST was not better than the one achieved in the four patients in that study who received palliative RT (180 days) [11]. Alonso-Miguel et al. described a median survival time close to our results (96 days); however, in this study a second antiangiogenic drug was administered to the patients [21].

As in the study by Souza et al. [10], our dogs experienced an improvement in clinical signs (pain, erythema, and firmness of the lesions) in at least one characteristic in most animals at day 15 (80%) and day 30 (66%) after starting the treatment. In other canine cancers, TKIs have improved tumor-related clinical signs, despite the progression of the disease according to RECIST criteria [39]. Therefore, TKIs could have an effect that targets the angiogenic component of the tumor through unknown mechanisms. In canine IMC, this could be due to antiangiogenic activity and/or blockade of VEGFR pathways implicated in lymphangiogenesis, an over-activated pathway in this disease, leading to an improvement in lymph flow and edema [8,36]. This clinical improvement was observed in all the dogs with lymphedema of the extremities in our study.

No dogs responded to the rescue treatments, which could have several explanations, including new studies where both canine IMC and human IBC present a higher expression of P-glycoprotein, potentially being partly the cause of the chemoresistance seen in this disease [40].

As with aggressive mammary tumors in humans, the treatment of canine IMC is primarily palliative. Thus, assessing the QOL via questionnaires and VAS measurements was a crucial goal of our study. To the authors’ knowledge, this approach has not been previously described in this disease in dogs. QOL assessment is a critical component in veterinary oncology, especially for aggressive cancer types like canine IMC. It provides valuable insights into the patient’s well-being, guides treatment decisions, and ensures comprehensive, patient-centered care [22,23]. In our study, QOL assessments included various parameters such as happiness, mental state, appetite, and general mobility. The results from the QOL questionnaires, including VAS evaluations, demonstrated significant improvements in several variables. Specifically, the VAS scores for QOL showed statistically significant improvements between diagnosis and 15- or 30-days post-treatment. Improvements were noted in areas such as happiness, mental status, appetite, and general mobility of the patients included in the study. However, it is important to note that the observed increase in mobility may have been influenced by the effects of carprofen on osteoarthritis rather than a direct antitumor effect [38]. This potential bias underscores the need for careful interpretation of QOL improvements, considering all factors that might contribute to the patient’s enhanced condition. Additionally, there is a potential recall bias from owners’ recollections of their pets’ symptoms and quality of life. Those with better outcomes might report more positively, while those with poorer outcomes might report more negatively. Our study explicitly used VAS and detailed QOL questionnaires to measure specific aspects of well-being (pain, mobility, appetite, happiness), while the other two studies [11,21] assessed CB through response rates and survival times but lacked detailed QOL metrics.

AEs were very mild and controlled with supportive treatment in our cohort of patients. As described previously, the most common side effects were gastrointestinal, including vomiting, diarrhea, and anorexia [25]. Grade three and four AEs were not recorded.

## 5. Conclusions

In conclusion, our study suggests that dogs with IMC treated with toceranib phosphate and carprofen as a first-line therapy can experience a CB (60%) with a toxicity profile similar to previous studies combining toceranib with a second metronomic agent. It also provides a focused evaluation of a simpler dual-drug regimen on QOL and clinical outcomes in dogs with IMC, using detailed QOL questionnaires to capture improvements in well-being. While other studies report higher CB rates and longer survival times, they use more complex treatment combinations including radiotherapy without any specific QOL metrics. The integration of detailed QOL assessments in future studies could enhance understanding of treatment impacts on patient well-being, aligning clinical outcomes with owner-reported benefits.

## Figures and Tables

**Figure 1 vetsci-11-00430-f001:**
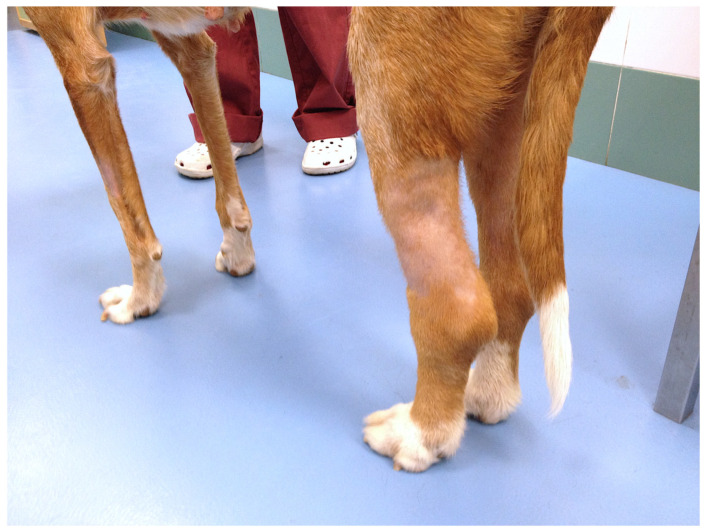
Lymphedema affecting both hindlimbs, slightly worse on the rear left leg.

**Figure 2 vetsci-11-00430-f002:**
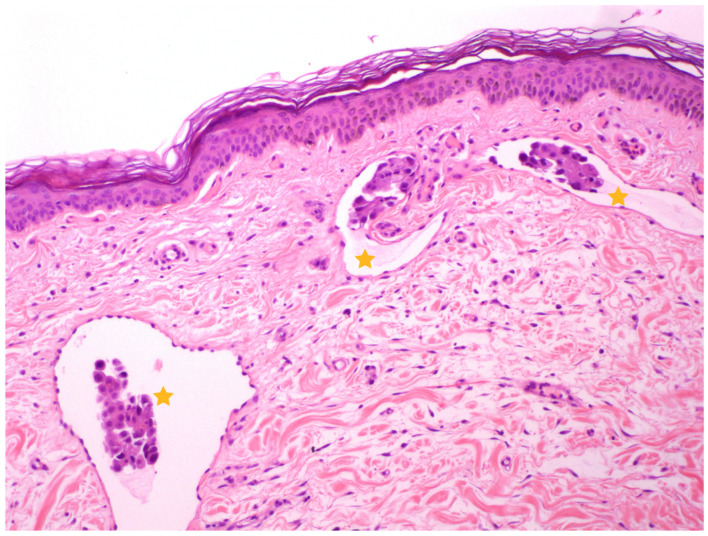
Sample of a skin histological section, stained with Hematoxylin and Eosin (10×). Lymphatic vessels in the superficial dermis are observed with intraluminal emboli of epithelial neoplastic cells (yellow stars), consistent with the diagnosis of IMC. Image courtesy of Carolina Naranjo, LV, PhD, DACVP, DECVP.

**Figure 3 vetsci-11-00430-f003:**
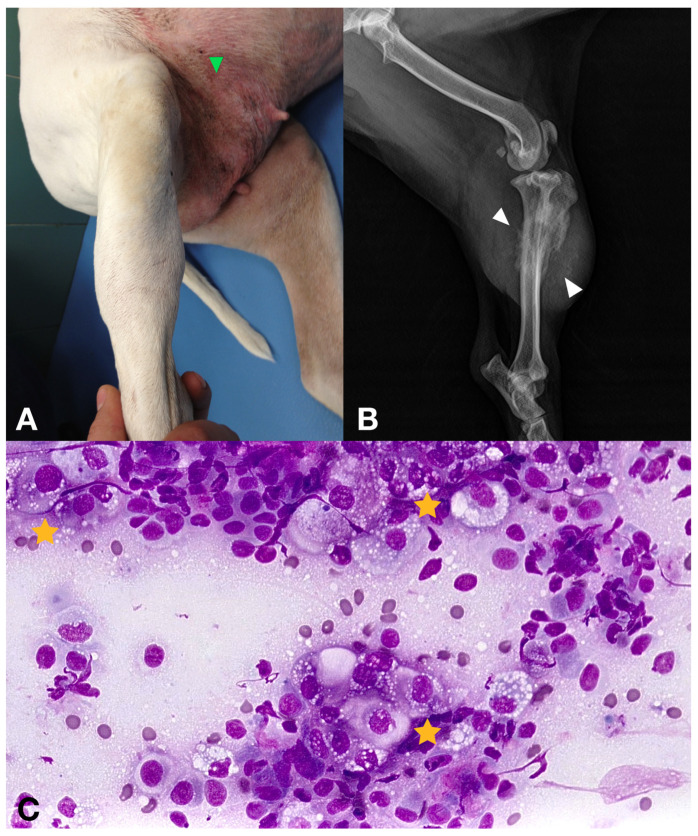
(**A**) 16-year-old female with a secondary IMC showing a diaphyseal tibial firm mass on the right rear leg. Erythema and inflammation with cutaneous nodules were also obvious in the inguinal mammary glands (green arrowhead). (**B**) Diaphyseal tibial aggressive bone lesion with osteolysis and osteoproliferation (white arrowheads). (**C**) Cytology obtained via 18 G needle aspirate under sedation. Epithelial cell clusters revealing atypical cells consistent with metastatic neoplasia (yellow stars).

**Figure 4 vetsci-11-00430-f004:**
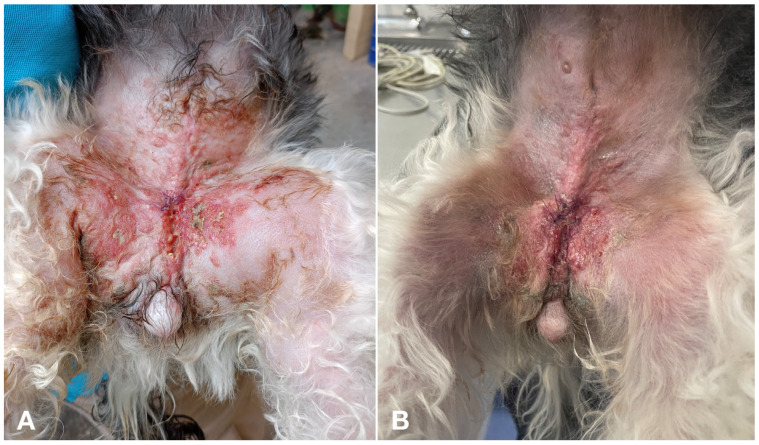
(**A**) Erythema and cutaneous nodules in a female dog with primary IMC. (**B**) The same dog as in image A, showing slight improvement of erythema and degree of inflammation after one month of treatment.

**Figure 5 vetsci-11-00430-f005:**
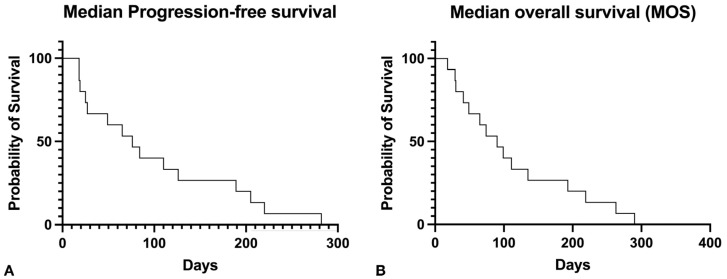
(**A**) Kaplan–Meier curve for progression-free survival. The median PFS was seventy-six days. (**B**) Kaplan–Meier curve for overall survival. Median overall survival was ninety days.

**Figure 6 vetsci-11-00430-f006:**
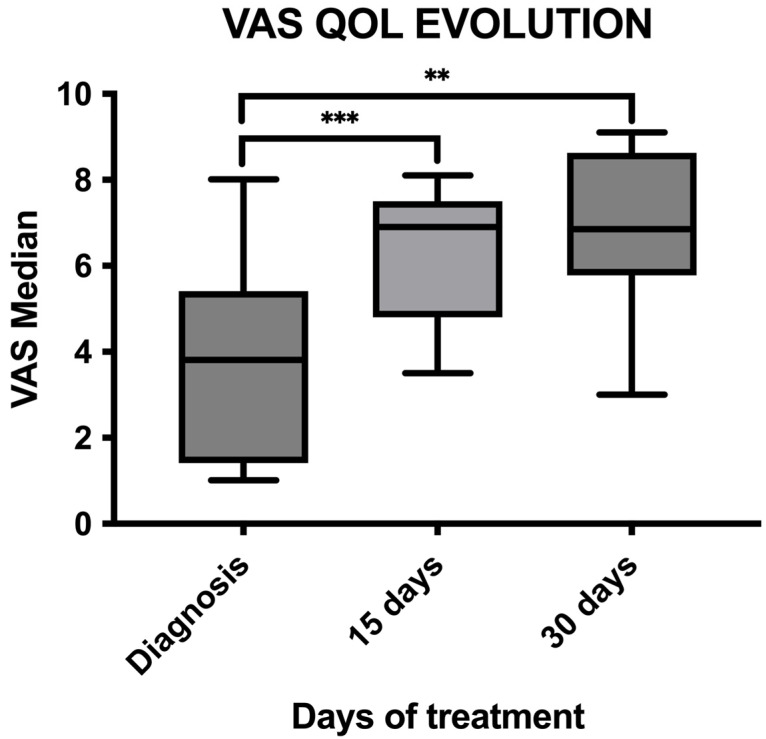
Graph comparing median QOL VAS between diagnosis (VAS median 3.66) and day 15 (VAS median 6.38), showing significant improvement (*p* = 0.0009), as well as between diagnosis and day 30 (VAS median 6.75), showing significant improvement (*p* = 0.001). No significant differences were found between day 15 and day 30 (*p* = 0.22). **, *p* < 0.01; ***, *p* < 0.001.

**Table 1 vetsci-11-00430-t001:** Tumors and cases characteristics.

Case	Age	Weight (kg)	Breed	Spayed	Years since OHE	Location	Other Characteristics	Size (cm)	Local or Distant Metastasis
1	9	15.5	Mixed	No	ND	L3–L5	Lymphedema Skin nodules	5.3	NDM
2	13	23.3	Mixed	No	ND	L3	Skin nodules	5.4	NDM
3	14	22.4	Mixed	Yes	10	R2–R3	Skin nodules	6.6	NDM
4	12	42.6	Mastín	Yes	11	R4–R5	Lymphedema Skin nodules	4.7	Iliac and inguinal lymph node
5	16	13.5	Mixed	No	ND	L4–L5	Skin	5.5	Diaphyseal tibial aggressive bone lesion
6	11	6.1	Yorkshire Terrier	No	ND	R3	Skin ulcers	3.4	NDM
7	8	3.3	Yorkshire Terrier	Yes	7	R1	Skin nodules	2.3	NDM
8	11	15.5	Mixed	No	ND	L5–R5	Lymphedema	6.7	Popliteal lymph node
9	13	42.7	Mixed	Yes	1	R1	Skin ulcers	5.6	NDM
10	12	9.1	French Bulldog	No	ND	L3–L4	Skin ulcers	4.6	NDM
11	8	27.8	Labrador	No	ND	L2–L3	Skin nodules	8.4	NDM
12	14	33.9	Rottweiller	Yes	5	R4	Skin nodules	9.6	Iliac lymph node
13	8	14.2	English Cocker spaniel	Yes	7	L2–L3	Skin nodules	3.4	NDM
14	15	28.6	German Shepherd	Yes	5	L4–L5	Lymphedema Skin nodules	8.7	Iliac and inguinal lymph node
15	10	29.9	Boxer	No	ND	R4	Skin nodules	6.7	Iliac and inguinal lymph node

OHE, ovariohysterectomy. ND, not determined. R, right mammary glands. L, left mammary glands. NDM, no distant metastasis.

**Table 2 vetsci-11-00430-t002:** The change in various variables from the quality-of-life questionnaire filled out by owners upon receiving the treatment. Scores range from 1 to 5, with 1 being “strongly disagree” and 5 being “strongly agree” with the given statement or question. Each value in the table represents the mean of the values from the 15 patients.

Topic	Question	Before Treatment	15 Days after	30 Days after
Happiness	My dog wants to play	3	3.867 * (*p* = 0.0039)	4.214 * (*p* = 0.049)
	My dog interacts with me	4.133	4.867 (*p* = 0.0078)	4.286 (*p* = 0.9)
	My dog is happy	3.571	4.467 * (*p* = 0.0039)	4.286 (*p* = 0.1250)
Mental status	My dog has more good days than bad days	3.071	4.2 * (*p* = 0.0010)	4.214 * (*p* = 0.0059)
	My dog is awake less	2.643	1.733 * (*p* = 0.0010)	2 (*p* = 0.12)
	My dog is depressed	2.5	1.667 * (*p* = 0.0010)	2.071 (*p* = 0.273)
Pain	My dog is in pain	4	2.267 * (*p* = 0.0002)	2.286 * (*p* = 0.0020)
Appetite	My dog eats all of their food portion	2.786	3.867 (*p* = 0.1675)	4.143 * (*p* = 0.0082)
	My dog is sick	1.071	1.4 (*p* = 0.1875)	1.143 (*p* = 0.9)
	My dog wants treats/snacks	3.286	4.133 (*p* = 0.12)	4.429 * (*p* = 0.0115)
Hydration	My dog drinks as normal	4.571	4.8 (*p* = 0.06)	4.643 (*p* = 0.9)
	My dog has diarrhea	1.286	1.533 (*p* = 0.5)	1.571 (*p* = 0.75)
	My dog urinates the same quantity as always	4.857	4.733 (*p* = 0.9)	4.714 (*p* = 0.75)
Mobility	My dog is moving normally	3.286	4 * (*p* = 0.0078)	4.143 * (*p* = 0.0215)
	My dog is resting all day	1.8	1.533 (*p* = 0.218)	1.643 (*p* = 0.9)
	My dog has higher activity than normal	2.143	3.133 * (*p* = 0.0020)	3.143 * (*p* = 0.0039)
General health	Health status compared to before the cancer diagnosis	2.33	4.27 * (*p* = 0.0001)	4.143 * (*p* = 0.0005)

The change in each value at 15 days and 30 days is compared with the pretreatment value using a paired *t*-test analysis. Significant changes, defined as those with *p* < 0.05, are marked with an asterisk (*).

## Data Availability

The data supporting the conclusions of this article will be made available by the authors, and the reasonable request.

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
