# Peer review of "Impact of Toceranib Phosphate and Carprofen on Survival and Quality of Life in Dogs with Inflammatory Mammary Carcinomas"

_vetsci, 2024, doi:10.3390/vetsci11090430_

Round 1

Reviewer 1 Report

Comments and Suggestions for Authors

This is a interesting study about the use of toceranb and carprofen on the palliative treatment of inflammatory mammary carcinoma in dogs. Despite the low number of cases the results are well presented and discussed. Minor revisions are suggested as follow:

Please state on the summaries that there was no partial or complete response, but stable disease, at 30 days, for 60% of patients. PFS should also be included in the short summary.

Line 54 - replace unfavorable for "poor".

On table 1 please include a column with information of distant metastasis (it is not necessary to add lymph node metastasis as evaluation was not posible in all cases). 

Comments on the Quality of English Language

There are few terms that should be replaced, as unfavorable prognosis to poor prognosis.

Author Response

Response to Reviewer 1 Comments

Thank you very much for taking the time to review this manuscript. Please find the detailed responses below and the corresponding corrections in track changes in the re-submitted manuscript.

Comments 1: Please state on the summaries that there was no partial or complete response, but stable disease, at 30 days, for 60% of patients. PFS should also be included in the short summary.

Response 1: Thank you for your valuable feedback. In response to your request, we have updated the summaries to clarify that the dogs did not exhibit a partial or complete response. Additionally, we have incorporated the progression-free survival (PFS) data into the short summary to provide a more comprehensive overview of the treatment outcomes. These changes can be found on page 1, lines 21, 23, and 24.

Comment 2: Line 54 - replace unfavorable for "poor".

Response 2: Thank you for your comment. We agree with this point and have made the necessary modifications to address it. The changes can be found on page 2, second paragraph, line 67.

Comments 3: On table 1 please include a column with information of distant metastasis (it is not necessary to add lymph node metastasis as evaluation was not possible in all cases). 

Response 3: Thank you for pointing this out. We agree with your suggestion and have added a column in Table 1 to specify whether metastasis was local or distant. Cases with no distant metastasis (NDM) are indicated where lymph nodes appeared normal based on physical examination or abdominal ultrasound, and thoracic radiographs showed no abnormalities. This change can be found on page 5, Table 1.

Comments on the Quality of English Language: There are few terms that should be replaced, as unfavorable prognosis to poor prognosis.

Response to Comments on the Quality of English Language We appreciate your feedback regarding the quality of the English language. As suggested, we have made the necessary adjustments to improve the clarity of the manuscript. The revisions can be found on page 2, second paragraph, line 67.

Thank you very much for your thoughtful and constructive feedback on our manuscript. We have carefully addressed all of your suggestions and believe the revisions have greatly improved the clarity and quality of the work. Your input has been invaluable, and I truly appreciate the time you have taken to help us enhance this paper.

If there is anything else you think could be further refined, please do not hesitate to let me know.

With sincere thanks,

Reviewer 2 Report

Comments and Suggestions for Authors

The topic is relevant to the veterinary literature and explore a very important condition of dogs. The manuscript is well written and presents promising results.  

A few considerations and questions are listed below: 

Simple summary 

Despite possible, consider avoiding the term ‘breast’ because it is usually applied to humans. Use of term 'mammary’ is suggested. 

Abstract 

Consider use of the term ‘mammary’ instead of ‘breast’. 

Authors should be cautious about overspeculation. Despite possible association with VEGF overexpression, inflammatory mammary cancer is not characterized by it. Consider rewriting the sentence (e.g., “... aggressive type of cancer in which vascular endothelial growth factor and cyclooxigenase-2 overexpression usually/may occur...”). 

INTRODUCTION 

Page 2: 

Line 46: Despite not common, inflammatory mammary cancer is not rare - reinforced by the subsequently presented percentage. Consider use of other terms (e.g., infrequent, uncommon, etc.). 

Line 54: Consider rewriting the sentence to make it more concise (e.g., “In women, inflammatory...”). 

Line 81: References should be included after ‘canine IMC’ (i.e., ...”canine IMC [18-20], the use of toracenib in this disease...”). 

In addition, authors could reinforce/highlight the innovative use of the selected drugs in the text and why it is important – especially in the third and fourth paragraphs. 

Page 3: 

MATERIALS AND METHODS 

Inform how the sample size was calculated or indicate why it was not calculated (e.g., retrospective/retroprospective study with all available cases during the period), as well as exclusion criteria. Also inform if consecutive or non-consecutive cases were evaluated. 

Line 113: Consider rewriting the sentence. It is more appropriate to say that tissue sections were obtained or that tissue/blocks were sectioned than cut – especially because tissue sections can be subsequently cut. 

Lines 116-118: Pointing that secondary IMC is associated with occurrence of disease after a previous diagnosis of other type of mammary tumor may imply in an inaccurate association with tumor histotypes. Since there is no specific histotype associated with inflammatory carcinomas, this may lead to misinterpretation. In this case, it is more appropriate to use the term ‘non-inflammatory mammary carcinoma/cancer’ (e.g., “...removal and diagnosis of non-inflammatory mammary cancer were classified as secondary.”). 

Line 130: Despite commonly used, Diff quik is not a stain but a commercial product/brand. Appropriate description of the method is based on the use of ‘rapid Romanowsky stain’ (Diff Quik, Merck....). 

Lines 130-132: Consider using ‘cell clusters’ instead of ‘clusters’. Inform used malignancy criteria (references) or indicate identification of ‘nuclear atypia’. Explain or indicate within the text why only cell clusters – and not individualized atypical epithelial cells – were considered for metastasis confirmation. 

RESULTS 

Page 5: 

Table 1. Standardize use of terms (lymphoedema/lymphedema). Also inform specific breed for Cocker Spaniel (English/American). 

Lines 209-210. As previously pointed, caution should be taken when talking about histopathological diagnosis. Inflammatory mammary cancer is a clinicopathological entity and not a histopathological diagnosis. There are histologic features that could be associated with the clinical entity and can be associated with clinical signs for diagnostic confirmation, but inflammatory cancer is not a tumor histotype. Consider using the term ‘non-inflammatory cancer/carcinoma’ diagnosis to avoid misinterpretation. 

Line 223: Consider use of “histopathological confirmation” instead of “histopathological diagnosis” since the latter term may lead to misinterpretation (histotype diagnosis). 

Figure 3. Consider including arrows/arrowheads/asterisks to indicate alterations described in the figure legend. Consider rewriting ‘C’ (e.g., Epithelial cell clusters revealing atypical cells consistent with metastatic neoplasia...) 

Table 2. The table is not necessary with data presented as it is since it is only replicating the exact same data immediately described above. Consider removing or including additional data/evaluations.  

DISCUSSION 

Clearly identified speculation can always add to the value of a scientific report. In this study, authors show that the applied questionnaire is promising. However, such aspect is not thoroughly discussed. Reinforcement of such feature to highlight the results and its benefits is suggested. Additionally, despite the few available cases, overfocusing in such fact in not compelling. There is also excessive focus on traditionally studied clinicopathologic features that should be discussed but are not the main objective of the study (treatment and quality of life assessment). Thus, authors could provide a wider discussion about the aforementioned aspects. 

Page 10: 

Lines 328-329: Provide references for the sentence after ‘angiogenic phenotype.’. 

Line 331: Consider use of ‘women’ or ‘humans’ instead of ‘human medicine’. 

Page 11: 

Lines 364-366: Consider rewriting as two separate sentences (e.g., “unmeasurable lesions. Similarly to humans, most canine...”). 

General considerations: 

The manuscript is well written, and the study shows potential but needs to be reviewed for further considerations – particularly the discussion section. There is some concern about ethical principles, especially when taking NC3Rs in consideration – https://nc3rs.org.uk/ .

Author Response

Response to Reviewer 2 Comments

Thank you very much for taking the time to review this manuscript. Please find the detailed responses below and the corresponding corrections in track changes in the re-submitted manuscript.

Comment 1: Simple summary: Despite possible, consider avoiding the term ‘breast’ because it is usually applied to humans. Use of term 'mammary’ is suggested. 

Response 1: Thank you for your suggestion to use the term 'mammary' instead of 'breast' in describing the condition in our research. We understand the distinction and the importance of using terminology that aligns more closely with veterinary contexts. Accordingly, we have made the necessary adjustments to ensure consistency and appropriateness throughout the manuscript, replacing 'breast' with 'mammary' where applicable. This change can be found on page 1, line 17.

Comment 2: Abstract: Consider use of the term ‘mammary’ instead of ‘breast’. 

Authors should be cautious about overspeculation. Despite possible association with VEGF overexpression, inflammatory mammary cancer is not characterized by it. Consider rewriting the sentence (e.g., “... aggressive type of cancer in which vascular endothelial growth factor and cyclooxigenase-2 overexpression usually/may occur...”). 

Response 2: Thank you for your feedback on our manuscript. We will incorporate the term 'mammary' where appropriate to align with the nature of the condition discussed. These changes have been made on page 1, lines 29-30.

Additionally, we appreciate your caution regarding overspeculation and the need for accuracy in our statements. We have revised the sentence to adopt a more cautious tone, indicating a potential association with VEGF overexpression in inflammatory mammary cancer without suggesting it as a definitive characteristic. This revision can be found on page 1, lines 30-31.

Comment 3: Line 46: Despite not common, inflammatory mammary cancer is not rare - reinforced by the subsequently presented percentage. Consider use of other terms (e.g., infrequent, uncommon, etc.). 

Response 3: Thank you for your feedback. We appreciate your observation that while inflammatory mammary cancer may not be considered common, it is also not rare, as indicated by the percentage data presented. In light of your suggestion, we have opted to use the term 'infrequent' on page 2, line 59, to more accurately describe the frequency of inflammatory mammary cancer in our text.

Comment 4: Line 54: Consider rewriting the sentence to make it more concise (e.g., “In women, inflammatory...”). 

Response 4: Thank you for your input. In response to your recommendation, we have revised the sentence to enhance conciseness while maintaining clarity and precision. The updated version can be found on page 2, line 67.

Comment 5: Line 81: References should be included after ‘canine IMC’ (i.e., ...”canine IMC [18-20], the use of toracenib in this disease...”). In addition, authors could reinforce/highlight the innovative use of the selected drugs in the text and why it is important – especially in the third and fourth paragraphs. 

Response 5: Thank you for your comment regarding Line 81. We have added references after "canine IMC" for clarity and proper citation. Additionally, we appreciate your suggestion to reinforce the innovative use of the selected drugs in the text. Page 2, line 93 and page 11, lines 417 – 421.

Comment 6: MATERIALS AND METHODS: Inform how the sample size was calculated or indicate why it was not calculated (e.g., retrospective/retroprospective study with all available cases during the period), as well as exclusion criteria. Also inform if consecutive or non-consecutive cases were evaluated. 

Response 6: Thank you for your feedback and for helping us improve the manuscript, particularly regarding the materials and methods section. As this study is retrospective, a formal sample size calculation was not conducted. Instead, we included all available cases managed by a single clinician, ensuring that only consecutive cases meeting the inclusion criteria were evaluated. To clarify this approach, we have added exclusion criteria to the methodology section. These changes can be found on page 3, lines 120-123, and page 4, lines 185-188.

Comment 7: Line 113: Consider rewriting the sentence. It is more appropriate to say that tissue sections were obtained or that tissue/blocks were sectioned than cut – especially because tissue sections can be subsequently cut. 

Response 7: Regarding line 113, we appreciate your suggestion to improve the clarity of the sentence. We will revise it to state that "tissue blocks were sectioned," ensuring that the terminology accurately reflects the process. Your input has been invaluable in enhancing the precision of our manuscript. The revision can be found on page 3, line 133.

Comment 8: Lines 116-118: Pointing that secondary IMC is associated with occurrence of disease after a previous diagnosis of other type of mammary tumor may imply in an inaccurate association with tumor histotypes. Since there is no specific histotype associated with inflammatory carcinomas, this may lead to misinterpretation. In this case, it is more appropriate to use the term ‘non-inflammatory mammary carcinoma/cancer’ (e.g., “...removal and diagnosis of non-inflammatory mammary cancer were classified as secondary.”). 

Response 8: Thank you for your comment. We understand your concern regarding the potential for misinterpretation. In response, we will revise the text to use the term "non-inflammatory mammary carcinoma," as you suggested. We appreciate your feedback and have made this change on page 3, lines 137-139.

Comment 9: Line 130: Despite commonly used, Diff quik is not a stain but a commercial product/brand. Appropriate description of the method is based on the use of ‘rapid Romanowsky stain’ (Diff Quik, Merck....). 

Response 9: Regarding line 130, we appreciate your clarification regarding the terminology. We will revise the sentence to accurately describe the method using "quick panoptic stain (Hemacolor, Merck, Darmstadt, Germany)." This change can be found on page 3, line 151.

Comment 10: Lines 130-132: Consider using ‘cell clusters’ instead of ‘clusters’. Inform used malignancy criteria (references) or indicate identification of ‘nuclear atypia’. Explain or indicate within the text why only cell clusters – and not individualized atypical epithelial cells – were considered for metastasis confirmation. 

Response 10:  Thank you for highlighting the issues in lines 130-132. We will revise the text to use the term "atypical epithelial cell clusters" for greater clarity. Additionally, we have rewritten the specific phrase from “malignancy criteria” to “nuclear atypia.” Furthermore, we have specified that individual atypical epithelial cells or atypical epithelial cell clusters were considered when confirming metastasis. These changes are reflected on page 3, line 152. 

Comment 11: Table 1. Standardize use of terms (lymphoedema/lymphedema). Also inform specific breed for Cocker Spaniel (English/American). 

Response 11: Thank you for your comment regarding Table 1. We have standardized the terms "lymphedema" and "English Cocker Spaniel" as suggested. These revisions will help enhance the clarity and precision of our manuscript. Please see the changes on page 5, Table 1.

Comment 12: Lines 209-210. As previously pointed, caution should be taken when talking about histopathological diagnosis. Inflammatory mammary cancer is a clinicopathological entity and not a histopathological diagnosis. There are histologic features that could be associated with the clinical entity and can be associated with clinical signs for diagnostic confirmation, but inflammatory cancer is not a tumor histotype. Consider using the term ‘non-inflammatory cancer/carcinoma’ diagnosis to avoid misinterpretation. 

Response 12: Thank you for your input regarding lines 209-210. We have revised the text to use "non-inflammatory carcinoma" instead of referring to histotype. This change clarifies the nature of the diagnosis and better reflects the relationship between histological features and clinical signs. The revision can be found on page 6, line 273.

Comment 13: Line 223: Consider use of “histopathological confirmation” instead of “histopathological diagnosis” since the latter term may lead to misinterpretation (histotype diagnosis). 

Response 13: Thank you for your suggestion regarding line 223. We have revised the term to "histopathological confirmation" to prevent any potential misinterpretation. This change can be found on page 6, line 286.

Comment 14: Figure 3. Consider including arrows/arrowheads/asterisks to indicate alterations described in the figure legend. Consider rewriting ‘C’ (e.g., Epithelial cell clusters revealing atypical cells consistent with metastatic neoplasia...) 

Response 14: Thank you for your suggestions regarding Figure 3. We agree that these changes will enhance our manuscript. We have added arrowheads and stars to improve clarity. Additionally, we have revised the description to: “Epithelial cell clusters revealing atypical cells consistent with metastatic neoplasia.” These updates can be found on pages 7-8, Figure 3.

Comment 15: Table 2. The table is not necessary with data presented as it is since it is only replicating the exact same data immediately described above. Consider removing or including additional data/evaluations.  

Response 15: Thank you for your comment regarding Table 2. We agree that the table is redundant as it replicates data already described in the text. Therefore, we will remove the table to streamline the presentation of our findings.

Comment 16: Clearly identified speculation can always add to the value of a scientific report. In this study, authors show that the applied questionnaire is promising. However, such aspect is not thoroughly discussed. Reinforcement of such feature to highlight the results and its benefits is suggested. Additionally, despite the few available cases, overfocusing in such fact in not compelling. There is also excessive focus on traditionally studied clinicopathologic features that should be discussed but are not the main objective of the study (treatment and quality of life assessment). Thus, authors could provide a wider discussion about the aforementioned aspects. 

Response 16: Thank you for your insightful comment regarding the need to further discuss the convenience of this regimen. We have expanded this discussion in the manuscript to highlight these important factors, focusing on how the regimen not only addresses clinical and quality of life (QoL) outcomes but also offers a more accessible and economically feasible option for managing canine IMC. This change can be found on page 11, lines 417-421.

Comment 17: Lines 328-329: Provide references for the sentence after ‘angiogenic phenotype.’. 

Response 17: We have added a reference to support the use of the term “angiogenic phenotype.” This addition can be found on page 10, line 405.

Comment 18: Line 331: Consider use of ‘women’ or ‘humans’ instead of ‘human medicine’. 

Response 18: Thank you for your suggestion regarding line 331. We have revised the text to use "women" instead of "human medicine." This change can be found on page 10, line 403.

Comment 19: Lines 364-366: Consider rewriting as two separate sentences (e.g., “unmeasurable lesions. Similarly to humans, most canine...”). 

Response 19: Regarding lines 364-366, we have rewritten the text to separate the content into two distinct sentences for improved clarity. The revised text can be found on page 11, line 404.

Comment 20:The manuscript is well written, and the study shows potential but needs to be reviewed for further considerations – particularly the discussion section. There is some concern about ethical principles, especially when taking NC3Rs in consideration – https://nc3rs.org.uk/ .

Response 20: Thank you for highlighting the importance of ethical considerations, particularly in light of the NC3Rs principles (Replacement, Refinement, and Reduction). We appreciate your concern and fully recognize the need to minimize animal suffering and prioritize the use of alternatives where possible.

In designing our study, we adhered to Good Clinical Practice (GCP) standards, even though they were not mandatory due to the retrospective nature of our research. We consistently obtained informed consent from pet owners and provided them with detailed information about the study. Additionally, our primary objective was to enhance the quality of life of our patients, which we assessed through additional questionnaires provided to owners. We believe this approach aligns well with the NC3Rs framework:

  1. Replacement: Our study utilized existing data from client-owned animals that had already received treatment in a clinical setting. This retrospective approach avoids the need for new animal subjects, aligning with the Replacement principle. Moreover, all animals included in our study were treated with the informed consent of their owners, who were aware that their pets' clinical data could be used for future research.
  2. Refinement: The treatment protocols employed in our study were chosen to minimize distress and suffering. We selected a combination therapy of Palladia and carprofen for its potential clinical benefits and because it is less invasive and stressful compared to other options like radiotherapy or intravenous chemotherapy, which require additional interventions.
  3. Reduction: The retrospective nature of our study inherently supports the Reduction principle by maximizing the use of data from a limited number of animals. This reduces the need for additional animals in future studies.

We are committed to maintaining the highest ethical standards in our research. We hope this explanation addresses your concerns and clarifies how our study aligns with the NC3Rs principles.

Thank you for your constructive feedback and for drawing our attention to this important issue.

Thank you very much for your thoughtful and constructive feedback on our manuscript. We have carefully addressed all of your suggestions and believe the revisions have greatly improved the clarity and quality of the work. Your input has been invaluable, and I truly appreciate the time you have taken to help us enhance this paper.

If there is anything else you think could be further refined, please do not hesitate to let me know.

With sincere thanks,

Reviewer 3 Report

Comments and Suggestions for Authors

In this study, the authors assessed the impact of a palliative treatment on the tumor and quality of life in dogs with inflammatory mammary cancer.

General comment:

The merit of this study is to evaluate a simple palliative treatment of IMC in dogs and to measure its impact on both efficacy criteria and quality of life.

The low frequency and absence of standard of care make it difficult to run comparative trials. Comparison with other trials is therefore subject to potential biases due to criteria of enrollment, method of follow-up, evaluation parameters… It is a limitation of the study.

When comparing palliative treatments, comparison should not be limited to the efficacy and impact on QoL but also to the convenience and cost of the treatment for the pet owner.

More detailed comments:

Materials and Methods

-          Line 102: the cases were recruited between Dec 2012 and Dec 2015. By curiosity, why did the authors wait so long to publish those results?

Results

-          Line 314 (section 3.5): it is not clear when the rescue treatments were given and how they could interfere with the follow-up parameters reported in the study (QOL, median PFS, median OS).

Discussion

-          Line 340: even if it cannot be considered as a standard treatment, it would have been possible to compare with COX-2 inhibitor monotherapy. Why was it not done?

-          Lines 372–386: in this study, the authors included only measurable tumors. The authors should comment on this selection criteria and to which extent it could introduce a bias. According to Alonso-Miguel and co (ref 21), OST might be different depending on the macroscopic aspect of the tumor (“The presence of one or more mammary nodules, in contrast to patients which had only plaque-like lesions, was associated with shorter OST”).

-          Line 376: the authors indeed observed “a higher rate of SD” compared to Rossi’s study. However, with the limited number of animals enrolled in the studies and the different criteria of inclusion, one cannot draw the conclusion that one treatment is better that the other. This comment applies also to the statement on lines 382-384. The authors should then revisit their statements on the treatment comparisons.

-          Lines 459–480: the absence of standard of care and the low frequency of IMC make it difficult to run comparative studies. However, the absence of another group (placebo or SoC) makes it difficult to assess which fraction of VAS QOL score improvement is due to treatment or to placebo effect.

Conclusions:

-          Lines 490-491: the authors have completed “a focused evaluation of a simpler dual-drug regimen on QOL and clinical outcomes in dogs with IMC”. It is indeed an important aspect of the study which should be more extensively discussed. In the discussion, the authors emphasized the impact of their palliative treatment on RECIST and QoL criteria and compared this performance to other treatments. However, they did not discuss about the convenience and costs aspects. These are important criteria for palliative treatments. Avoiding chemotherapy or radiotherapy may facilitate the access to treatment for some patients. This should be developed a little more in the discussion.

Author Response

Response to Reviewer 3 Comments

Thank you very much for taking the time to review this manuscript. Please find the detailed responses below and the corresponding corrections in track changes in the re-submitted manuscript.

Comment 1: Line 102: the cases were recruited between Dec 2012 and Dec 2015. By curiosity, why did the authors wait so long to publish those results?

Response 1: Thank you for raising this question. The primary reason for the delay in writing up some of the studies is that the lead author, who initially conceptualized and conducted the study, transitioned from an academic institution to co-found a referral-only hospital while also starting a family. These changes limited the resources and time available to publish studies conducted during the time expend in academia. However, this situation has improved over the last couple of years, and we believe that the data collected, despite being older, still contributes valuable insights to the current body of knowledge on this type of tumor. Additionally, our first resident is expected to complete his training in 2024, which has motivated us to finalize these projects. 

Comment 2: Line 314 (section 3.5): it is not clear when the rescue treatments were given and how they could interfere with the follow-up parameters reported in the study (QOL, median PFS, median OS).

Response 2: Thank you for your comment regarding Line 314 in Section 3.5. We appreciate the importance of clarifying the timing of rescue treatments and their potential impact on the follow-up parameters reported in the study. In our manuscript, we have specified that rescue treatments were administered only after the occurrence of progressive disease. Therefore, rescue protocols did not affect the specific median progression-free survival (PFS) for Palladia.

While rescue protocols can influence overall survival (OS) and introduce bias—since more proactive owners may choose additional treatments, potentially extending individual outcomes artificially—in this study, only one animal received more than one dose of doxorubicin, which did not significantly affect OS. This clarification has been made on page 10, line 390.

Comment 3: Line 340: even if it cannot be considered as a standard treatment, it would have been possible to compare with COX-2 inhibitor monotherapy. Why was it not done?

Response 3: We apologize for any confusion, but we do not fully understand the question. We did not include a group treated solely with NSAIDs as a control because this was a retrospective study, and we lacked patients treated with NSAIDs alone at our institution. If the suggestion is to compare our results with the earlier studies by Clemente et al. 2009 [9], where small groups received only NSAIDs, we can certainly add that to the discussion. However, it is worth noting that in one of their control groups, which consisted of 23 cases treated only with NSAIDs, the overall survival time (OST) was 35 days. This is why we chose to compare our results with other combination protocols instead.

Comment 4: Lines 372–386: in this study, the authors included only measurable tumors. The authors should comment on this selection criteria and to which extent it could introduce a bias. According to Alonso-Miguel and co (ref 21), OST might be different depending on the macroscopic aspect of the tumor (“The presence of one or more mammary nodules, in contrast to patients which had only plaque-like lesions, was associated with shorter OST”).

Response 4: We appreciate your insightful comment regarding our selection criteria, which focused exclusively on measurable tumors in our study. In our experience, plaque-like lesions are challenging to measure according to RECIST standards, which could have introduced bias in our study if nodular lesions indeed have a worse prognosis than plaque-like lesions. However, both our study and that of Alonso-Miguel are small, making it difficult to definitively associate macroscopic characteristics with outcomes in their paper. We have expanded the discussion to emphasize the differences in our population characteristics, which includes fewer plaque-like lesions. These modifications can be found on page 11, lines 446-449.

Comment 5: Line 376: the authors indeed observed “a higher rate of SD” compared to Rossi’s study. However, with the limited number of animals enrolled in the studies and the different criteria of inclusion, one cannot draw the conclusion that one treatment is better that the other. This comment applies also to the statement on lines 382-384. The authors should then revisit their statements on the treatment comparisons.

Response 5: Thank you for your feedback regarding our statements on lines 376 and 382-384. We appreciate your point concerning the limitations posed by the small sample size and the differing inclusion criteria in our study compared to Rossi's.

In response, we have revised our language to state that we observed a "similar and comparable rate of stable disease (SD)" rather than suggesting a higher rate. Additionally, we have adjusted our discussion of treatment comparisons to reflect a more cautious interpretation of the results, acknowledging the limitations you highlighted. These changes can be found on page 11, lines 463 and 503.

Comment 6: Lines 459–480: the absence of standard of care and the low frequency of IMC make it difficult to run comparative studies. However, the absence of another group (placebo or SoC) makes it difficult to assess which fraction of VAS QOL score improvement is due to treatment or to placebo effect.

Response 6:

Thank you for your thoughtful comments regarding the inclusion of a placebo group or a standard-of-care control group receiving only NSAIDs in our study. We would like to clarify that, due to the retrospective nature of our research, it was not possible to implement these specific control groups.

Additionally, looking ahead to future prospective studies, we believe that using a placebo control group, which would withhold potentially beneficial treatment, could be ethically problematic in a canine study. While some studies include placebo controls during an initial inclusion phase and then offer the combination protocol, drug of interest, or standard-of-care group shortly thereafter—sometimes providing financial support to encourage enrollment—such an approach would be extremely difficult to implement in a disease as aggressive as this one.

In terms of including a historical control group within our institution, NSAID monotherapy was not a common practice. This decision was informed by prior literature, including the study by Clemente et al. [6], which reported a median survival time of only 35 days with NSAID monotherapy. Consequently, we typically opted for a combination approach to potentially improve patient outcomes, in line with current best practices and ethical considerations.

We hope this explanation clarifies our methodology and decisions. Please let us know if there are any further questions or suggestions.

.

Comment 7: Lines 490-491: the authors have completed “a focused evaluation of a simpler dual-drug regimen on QOL and clinical outcomes in dogs with IMC”. It is indeed an important aspect of the study which should be more extensively discussed. In the discussion, the authors emphasized the impact of their palliative treatment on RECIST and QoL criteria and compared this performance to other treatments. However, they did not discuss about the convenience and costs aspects. These are important criteria for palliative treatments. Avoiding chemotherapy or radiotherapy may facilitate the access to treatment for some patients. This should be developed a little more in the discussion.

Response 7: Thank you for your insightful comment regarding the need to further discuss the convenience and cost aspects of this regimen. We have expanded this discussion in the manuscript to highlight these important factors, demonstrating how the regimen not only addresses clinical and quality of life (QoL) outcomes but also offers a more accessible and economically feasible option for managing canine IMC. This change can be found on page 11, lines 417-421.

Thank you very much for your thoughtful and constructive feedback on our manuscript. We have carefully addressed all of your suggestions and believe the revisions have greatly improved the clarity and quality of the work. Your input has been invaluable, and I truly appreciate the time you have taken to help us enhance this paper.

If there is anything else you think could be further refined, please do not hesitate to let me know.

With sincere thanks,

Round 2

Reviewer 2 Report

Comments and Suggestions for Authors

The manuscript was considerably improved and the authors showed commitment to the review. Text is now more clear and concise.

Comments on the Quality of English Language

Text is well written. There is need for minor language edition.